# The Influence of Motivators and Barriers to Exercise on Attaining Physical Activity and Sedentary Time Guidelines among Canadian Undergraduate Students

**DOI:** 10.3390/ijerph191912225

**Published:** 2022-09-27

**Authors:** Liam P. Pellerine, Nick W. Bray, Jonathon R. Fowles, Joyla A. Furlano, Anisa Morava, Taniya S. Nagpal, Myles W. O’Brien

**Affiliations:** 1Division of Kinesiology, School of Health and Human Performance, Faculty of Health, Dalhousie University, Halifax, NS B3H 4R2, Canada; 2Department of Physiology & Pharmacology, Cumming School of Medicine, University of Calgary, Calgary, AB T2N 1N4, Canada; 3Centre of Lifestyle Studies, School of Kinesiology, Acadia University, Wolfville, NS B4P 2R6, Canada; 4Faculty of Health Sciences, McMaster University, Hamilton, ON L8N 3Z5, Canada; 5School of Kinesiology, University of Western Ontario, London, ON N6A 3K7, Canada; 6Faculty of Kinesiology Sport and Recreation, University of Alberta, Edmonton, AB T6G 2R3, Canada

**Keywords:** exercise motivations, university students, self-report survey, logistic regression analysis, perceptions of movement

## Abstract

Canadian 24 h movement guidelines recommend engaging in >150 min/week of moderate–vigorous-intensity physical activity and ≤8 h/day of sedentary time. Half of Canadian post-secondary students do not meet physical activity or sedentary time guidelines. This pan-Canadian study aimed to (1) identify commonly cited motivators/barriers to exercise, and (2) determine which motivators/barriers were most influential for attaining physical and sedentary activity guidelines. A total of 341 respondents (279 females, 23 ± 4 years old, 53% met activity guidelines, 49% met sedentary guidelines) completed an online survey regarding undergraduate student lifestyle behaviours. Improved physical health (74% of respondents), mental health (67%), physical appearance (60%), and athletic performance (28%) were the most common motivators to exercise. The most common barriers were school obligations (68%), time commitments (58%), job obligations (32%), and lack of available fitness classes (26%). Students citing improved athletic performance (odds ratio (OR) = 1.94, *p* = 0.02) were more likely to adhere to activity guidelines, while those who selected physical health (OR = 0.56, *p* = 0.03) and physical appearance (OR = 0.46, *p* = 0.001) as motivators were less likely to meet activity guidelines. Students who cited school obligations as a barrier were less likely (OR = 0.59, *p* = 0.03) to meet sedentary guidelines. The motivators and barriers identified provide a foundation for university-led initiatives aimed at promoting physical activity and reducing sedentary time among undergraduate students. Strategies that positively re-frame students’ physical health and appearance-based motivations for exercise may be particularly useful in helping more students achieve national activity recommendations.

## 1. Introduction

It is well established that engaging in physical activity and limiting sedentary time are associated with improved mental, cognitive, and physical health [1]. Canada’s 24-h movement guidelines recommend 150 min·week^−1^ of moderate-to-vigorous intensity aerobic physical activity (MVPA) and limiting sedentary (sitting, lying, reclining postures) time to <8 h·day^−1^ [2]. MVPA and sedentary time are independent lifestyle factors, meaning it is possible to be physically active but live a sedentary lifestyle [3]. Adherence to both guidelines is associated with a reduced risk of all-cause mortality, disease, and mental health issues across the lifespan [4]. However, 50% and 44% of post-secondary students do not meet the physical activity [5] and sedentary time guidelines [6], respectively. The daily lives of most post-secondary students have drastically changed throughout the COVID-19 pandemic, including spending more time at home, less social interaction, and suspension of on-campus activities [7]. As Canadian institutions transition back to on-campus learning, it is important to consider how the lifestyle behaviours of students have changed. This includes investigating the current physical activity and sedentary patterns of post-secondary students to help institutions adapt to “living with COVID-19”. We sought to identify how motivators and barriers to exercise have impacted lifestyle behaviours throughout the pandemic.

Behaviour change is often achieved when the motivation to do such behaviour exceeds the barriers that discourage it [8]. The self-determination theory demonstrates that perceived behavioural control and motives are independent of perceived barriers to exercise [9]. A review of the relationship between the self-determination theory and exercise found that people who exercise for intrinsic reasons (e.g., personal enjoyment) were more likely to exercise than those who are motivated extrinsically (e.g., body image, maintaining health) [10]. While motivators and barriers to exercising in post-secondary students have been investigated in individual Canadian colleges/universities [5,11], there is limited evidence regarding their influence on guideline adherence in nationwide samples of undergraduate students [12,13]. For college students across the United States, common motivators for exercising were ranked as improving general health, body image, stress reduction, and personal enjoyment, while the common barriers were ranked as lack of time, laziness/no interest, other obligations, and lack of motivation [13]. In a study of undergraduate students at the University of Toronto, students ranked lack of peer support as the most common barrier to physical activity, followed by social intimidation and poor time management [12]. More recently, graduate students at the University of Western Ontario ranked improving physical health, mental health, and challenging themselves as the biggest motivators for engaging in exercise, while the most common barriers were ranked as time commitments, lack of exercise classes, and research obligations [11]. While existing literature has described motivators and barriers to exercise, no previous study has quantified how motivators and barriers impact adherence to physical activity and sedentary time guidelines. Furthermore, being limited to individual institutions, it is difficult to externalize previous findings as geographic location, campus facilities, and campus programming may play a role in the types of motivators and barriers faced. Accordingly, studying a nationwide sample of Canadian undergraduate students to quantify both physical activity and sedentary time guidelines, and the motivators and barriers to achieving recommendations is needed.

The purpose of this study was to (1) report the most common motivators and barriers to exercise among Canadian undergraduates, and (2) determine which common motivators and barriers influenced the odds of Canadian undergraduate students meeting physical activity and sedentary time guidelines. Based on previous literature, it was hypothesized that the most reported motivators would be improvements to physical and mental health, while the most reported barriers would be time commitment, lack of interest, and lack of exercise classes/resources. 

## 2. Methods

### 2.1. Participants

A close-ended survey assessing the lifestyle behaviours of Canadian post-secondary undergraduate students was developed and endorsed by Exercise is Medicine Canada committee members. Exercise is Medicine Canada is a nationwide organization committed to promoting healthy active lifestyles for all Canadians. Through on-campus initiatives across the country, Exercise is Medicine Canada aims to improve campus facilities, programming, and resources to help students lead healthy lifestyles. For this study, questions assessing the physical activity and sedentary time levels of Canadian undergraduate students, including motivators and barriers to engaging in exercise, were included. The survey was open to any undergraduate student enrolled at a Canadian post-secondary institute at the time the survey was distributed. Participants were recruited through Exercise is Medicine Canada’s network of post-secondary institutions, which shared survey materials with undergraduate students via email, newsletters, and word-of-mouth. All survey responses were collected voluntarily and anonymously through a secure online survey platform (Qualtrics, Provo, UT, USA). Prior to beginning the survey, all participants were provided with a detailed overview of the study and virtual informed consent was obtained. Refer to Appendix A for a breakdown of the geographic locations of respondents’ institutions. The survey was launched from December 2021 to May 2022. Research ethics board approval was granted by Acadia University.

### 2.2. Survey

The survey was developed with multiple-choice, scalar, and ranking question types. The survey was released during the COVID-19 pandemic. Therefore, government restrictions varied from province to province. Participants were explicitly told, “this survey is assessing exercise behaviours during the Fall 2021 and Winter 2022 semesters”, and to respond accordingly. Survey questions were modelled based on existing questionnaires, including questions from the Physical Activity and Sedentary Behaviour Questionnaire (PASB-Q) [13]. Weekly physical activity levels were estimated via the Physical Activity Vital Sign (sub-section of PASB-Q), which is calculated by multiplying the answers of the following two questions: (1)“In a typical week, how many days do/did you do moderate-intensity (like brisk walking) to vigorous-intensity (like running) aerobic physical activity?”(2)“On average for days that you do/did at least moderate-intensity aerobic physical activity (as specified just above), how many minutes do/did you do?”

Adherence to the physical activity guidelines was determined by comparing the estimated value to the >150 min·week^−1^ threshold. Sedentary activity levels were estimated by calculating a weighted average [(5 × weekday + 2 × weekend)/7] using the following questions integrated from the PASB-Q: (1)“How many hours per day do you typically spend sitting, reclining, or lying down on a weekday? (Include time at work, school, at home or while commuting. Exclude time spent sleeping or napping).”(2)“How many hours per day do you typically spend sitting, reclining, or lying down on a weekend day? (Include time at work, school, at home or while commuting. Exclude time spent sleeping or napping).”

Adherence to the sedentary time guidelines was determined by comparing the estimated value to the <8 h per day threshold. Meeting the guidelines was coded as a value of “1” (yes) and not meeting was coded as “0” (no).

Participants could skip and not answer any question by using a “prefer not to disclose” option. For this study, questions were analysed with reference to undergraduate student population descriptors (e.g., age, sex, gender, year of study), physical activity levels, sedentary activity levels, and motivators and barriers to exercising. Respondent descriptors were summed and/or averaged across the sample.

All respondents were asked to rank their top three motivators for engaging in exercise out of nine predetermined options or to select “prefer not to disclose”:

“Please select the top three factors that motivate(d) you to exercise?”

They also selected their top three barriers preventing them from exercising out of 14 predetermined options or selected “prefer not to disclose”:

“Please select the top three barriers that prevent(ed) you from exercising?”

These questions and predetermined motivators/barriers were adapted from a previous study of Canadian graduate students [11]. For each motivator and barrier, the number of selected occurrences and percentage of the total sample were calculated. For continuous variables, ±3 standard deviation thresholds from the mean were used to identify the presence of outliers. Based on this, 52 individual data points (11 from age, 17 from body mass index (BMI), 6 from physical activity levels, and 18 from sedentary activity levels) were designated as outliers and excluded from analyses. 

### 2.3. Statistical Analyses

All statistical analyses were completed in IBM SPSS Statistics (Version 27) with a statistical significance threshold of α = 0.05. Data are presented as mean ± standard deviation. Logistic regression outcome variables were whether physical activity and sedentary time guidelines were met (dichotomous). The assumptions of binomial logistic regression requiring independence of data points, absence of strongly correlated predictor variables, and linearity of the variables to log odds were achieved (all variables in the model were dichotomous). To ensure the assumptions of complete information sampling (i.e., the sample includes every combination of yes/no predictors) and lack of overdispersion were achieved, the top four most commonly selected motivators and barriers were entered as predictors for each model. To check for sex and BMI for potential covariates, independent sample t-testing was used to compare between those who met and did not meet guidelines. The female portion of our sample was also examined independently for exploratory purposes. The top four motivators/barriers were entered into a four-block binomial logistic regression model, with the most popular predictor entering the model first and the fourth most popular predictor entering in the last block.

The significance of each logistic regression model was interpreted based on the *p*-value of the chi-squared test statistic for each block. If the model was significant, the *p*-values of the β-coefficients for each predictor variable were used to determine the most influential motivators and barriers. The exp(β) for each predictor provided the odds ratio of the predictor’s effect on the outcome occurring (i.e., the odds of meeting the guidelines). Forest plots were used to provide a summary of the odds ratios ±95% confidence intervals (asymmetrical error bars) for all the motivators and barriers. 

## 3. Results

A total of 341 of the 411 (83% completion rate) undergraduate respondents who began the survey completed the survey. Table 1 provides a summary of the sample characteristics (i.e., age, sex, gender, BMI, activity levels, geographic location of school, etc.). There were no significant differences for BMI and sex between guideline attainers and non-attainers (all *p* > 0.054). Additional sample information regarding the distribution of geographical location of school by province can be found in Appendix A. Across all schools, the top four selected motivators to exercise were improved physical health, mental health, physical appearance, and athletic performance (Table 2). The top four selected barriers to exercise were school obligations, time commitment, job obligations, and lack of fitness classes (Table 2). 

### 3.1. Adherence to Physical Activity Guidelines

The model predicting attainment to physical activity guidelines from motivators to exercise was statistically significant (χ^2^ = 28.0; *p* < 0.001). Improved athletic performance was a positive predictor (β = 0.66; *p* = 0.02; 1.9 times more likely) of physical activity guideline attainment, while improved physical appearance (β = −0.77; *p* < 0.001; 1.8 times less likely) and improved physical health (β = −0.59; *p* = 0.03; 1.3 times less likely) were negative predictors (Figure 1). When examining females only, improved athletic performance was a positive predictor (β = 0.863; *p* = 0.01; 2.4 times more likely), while improved physical appearance (β = −0.63; *p* = 0.016; 1.9 times less likely) was a negative predictor (Appendix A). Improved physical health (β = −0.60; *p* = 0.06; 1.8 times less likely) was no longer a predictor of physical activity guideline attainment (Appendix A). The model for the influence of barriers on physical activity guideline attainment was not significant (χ^2^ = 3.1; *p* = 0.54) and there were no barriers that were predictive of physical activity guideline adherence (Figure 1; all, *p* > 0.15). When considering females only, lack of fitness classes became a positive predictor (β = 0.59; *p* = 0.04; 1.8 times more likely) of physical activity guideline attainment.

### 3.2. Adherence to Sedentary Time Guidelines

The model of motivators’ influence on sedentary time guideline attainment was not significant (χ^2^ = 2.2; *p* = 0.69) and there were no motivators that were predictive of sedentary time guideline attainment (Figure 2; all, *p* > 0.063). The overall model associating barriers with sedentary time guideline attainment was significant (χ^2^ = 12.3; *p* = 0.02), with only school obligations (β = −0.53; *p* = 0.03; 1.7 times less likely) being a negative predictor of sedentary time guideline attainment (Figure 2). When considering females only, the results did not change for the motivator- and barrier-based sedentary guideline models (Appendix A).

## 4. Discussion

This study examined motivators and barriers to exercise in a Canadian cohort of undergraduate students, and the likelihood of these factors influencing adherence to national physical activity and sedentary time guidelines. The most cited motivators to exercise were improved physical health, mental health, physical appearance, and athletic performance, while the most common barriers to exercise were school obligations, time commitments, job obligations, and lack of fitness classes. These results support our hypotheses except for lack of students’ interest not being cited as a common barrier to exercise. We determined that being motivated by improved physical health or physical appearance reduced the odds of meeting physical activity guidelines, while motivation to improve athletic performance increased the odds of achieving physical activity guidelines. Undergraduates who cited school obligations as a barrier to exercising were less likely to achieve sedentary time guidelines. Altogether, the findings herein better our understanding of motivators and barriers to exercise among Canadian undergraduate students and may serve as variables of interest for health promotors to improve the lifestyle behaviours of this population.

Our respondents had similar physical activity guideline adherence (53%) to the rates observed across the undergraduate population (50%) [5]. When comparing the frequencies of motivators to exercise (Table 2) to previous research at a single institution, improved physical health and improved mental health remained as the most commonly reported [11]. Our findings further add that students who listed physical health as a motivator to exercise (74% of our sample) were 1.8 times less likely to meet the physical activity guidelines than those who did not cite this. This finding may suggest that being motivated to exercise to specifically improve your physical health may not be enough to achieve physical activity guidelines. This was even more apparent for respondents who were motivated to exercise to improve their physical appearance (60% of the sample), as they were 2.2 times less likely to meet physical activity guidelines. As reviewed elsewhere [9], there is mixed evidence on whether exercising to improve fitness and physical appearance is enough to self-motivate individuals to engage in exercise. Being primarily motivated by physical health and/or appearance may be viewed as something they know they should be doing, rather than something they want to be doing. For instance, students who exercise because they know it is important to maintain their health but possibly perceive the activity as a chore are likely less physically active than those who are engaging in exercise because they find it enjoyable [10]. Accordingly, it may be prudent to explore alternate health promotion strategies for encouraging movement beyond the focus on physical health and appearance. For example, focusing on psychosocial benefits like enjoyability and improved energy may be more optimal to increase adherence to physical activity guidelines [14].

Approximately one-quarter (28%) of undergraduate students were motivated by improved athletic performance, and those who cited it were 1.8 times more likely to achieve physical activity guidelines. Appendix A demonstrates that this result was not driven by varsity athletes, as only 36% of respondents motivated by athletic performance were varsity athletes. Students motivated by personal challenge and sports have been shown to meet activity guidelines more often [14]. Students who play both recreational and varsity sports are often held accountable by their teammates, coaches, and/or opponents to train, with these external factors facilitating a more physically active lifestyle [14]. Campuses may consider adding more recreational programs (e.g., running clubs, sports intramurals) to garner increased participation in physical activity [15]. It is also worth noting that this survey was completed by respondents while many institutions were not offering fitness classes or recreational sports due to COVID-19 restrictions. In the sub-analysis of our female sample (see Appendix A), we found that females respondents who listed “lack of fitness classes” as a barrier were more likely to attain to physical activity guidelines. It seems that many females have found alternatives to exercising despite a lack of fitness programming being available. Nevertheless, as universities transition back into on-campus learning, campuses should reintroduce fitness programs and invest more resources into facilitating these motivators and reducing barriers to exercise to boost activity levels of students. 

The proportion of students that met sedentary time guidelines (50%) in our study was similar to that in another national survey (56%) of Canadian post-secondary students [6]. Though half of our sample did not meet sedentary time guidelines, no common motivators to exercise significantly influenced their attainment to sedentary time guidelines. However, responders who cited school obligations as a barrier to exercise (68% of the sample) were 1.8 times less likely to meet the sedentary time guidelines. Limiting sedentary time has been shown to reduce risk of chronic disease (e.g., cardiovascular disease, diabetes, cancer) and all-cause mortality across one’s lifespan [16]. Students spend most of the school day sedentary, and this has increased when within the confines of the COVID-19 pandemic [7]. Almost all Canadian students had to adapt to an online learning format with on-campus activities being suspended, resulting in a major reduction in social interaction [17]. The lack of social interaction and increased confinement throughout the pandemic have been shown to increase sedentary time for students [7]. Interestingly, our study found respondents who reported COVID-19 restrictions met sedentary guidelines more often than those without COVID-19 restrictions (51% vs. 45%, respectively). Nevertheless, it is critical to implement strategies to reduce the sedentary time of undergraduate students. This may be achieved through on campus health promoting activities and social events that encourage movement. For example, the Exercise is Medicine Canada on Campus program organizes accessible student activities that encourage physical activity and reduction of sedentary time [18]. Previously hosted activities across Canada have included stretching breaks in class, advocating for standing desks, improving on-campus transportation (e.g., promoting walking/cycling paths), and adding more on-campus social events such as group fitness classes. 

A strength of this study was the use of a nationwide sample of undergraduate students to identify motivators and barriers to exercise rather than a single institution. Our sample may not be representative of the Canadian undergraduate population as a whole, with most respondents being cis-gendered, female students, studying in health disciplines. This may have led to higher activity levels and different motivator/barrier selections due to their increased health knowledge. Therefore, our findings cannot be generalized to represent the entire Canadian undergraduate population. Our survey had an undergraduate response rate of 83% (341 of the 411 viewers completed the survey). Furthermore, our findings add to the literature by exploring whether the identified motivators/barriers predicted adherence to guidelines. An individual’s motivation to exercise may be complex and influenced differently by specific behaviours and constructs; however, we aimed to determine the most impactful motivators and barriers to engaging in exercise. Interviews and/or focus groups would have permitted a more in-depth perspective on motivators/barriers to exercise but were not feasible for this study. The study may also be limited by its use of a self-reported questionnaire instead of objective measures that are more accurate for determining exact time spent in MVPA and in sedentary postures. However, broad dichotomizations of “met” versus “did not meet” guidelines were implemented to avoid the errors of self-report measures on exact levels of habitual activity. 

## 5. Conclusions

Canadian undergraduate students identified motivators and barriers to meeting national physical activity and sedentary time guidelines during the global pandemic. Focusing on physical health and appearance related motivators to exercise appeared to not be effective for attaining national guidelines. These findings can be used to inform health promotion activities on university campuses to encourage physical activity and reduce sedentary time, using population-informed motivators and barriers to guide initiatives.

## Figures and Tables

**Figure 1 ijerph-19-12225-f001:**
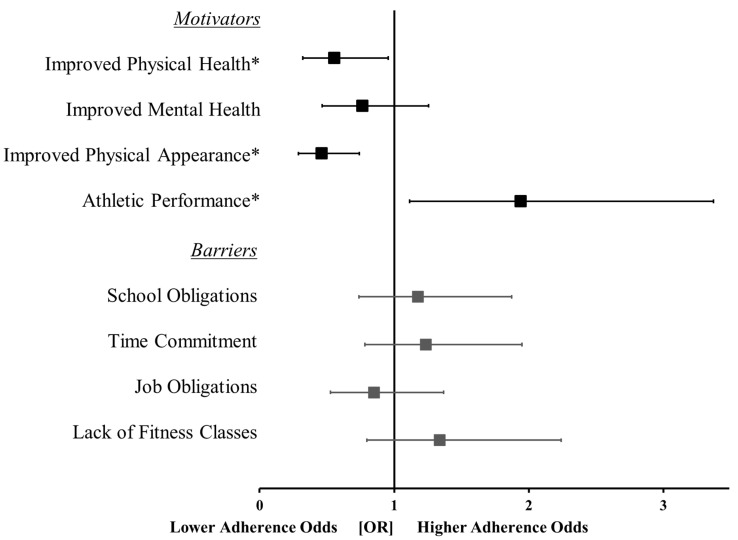
Forest plot displaying the odds ratios (OR) ± 95% confidence intervals for the motivators and barriers to achieving the physical activity guidelines. *, indicates a significance of *p* < 0.05. Note: Due to odds ratios having a minimum value of 0 (i.e., cannot be less than 0) and an infinite maximum value, the 95% confidence intervals were asymmetrical in magnitude.

**Figure 2 ijerph-19-12225-f002:**
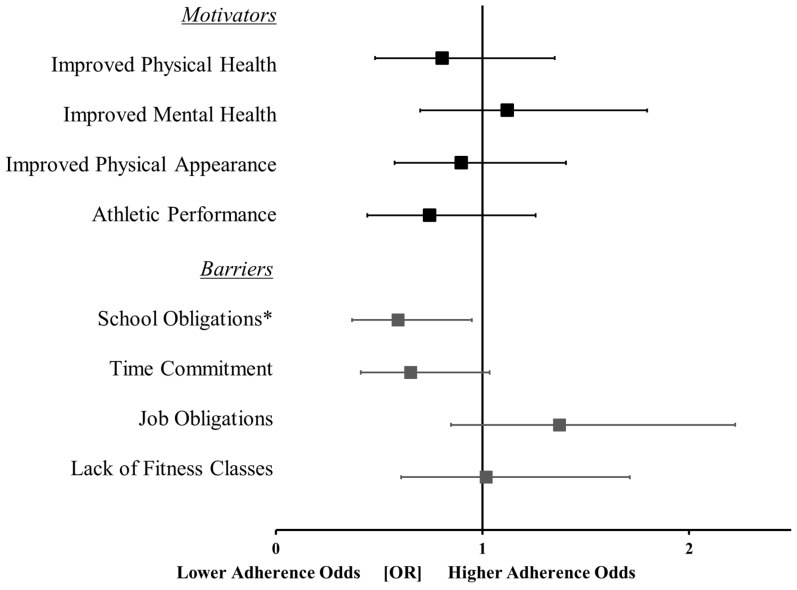
Forest plot displaying the odds ratios (OR) ± 95% confidence intervals for the motivators and barriers to achieving sedentary time guidelines. *, indicates a significance of *p* < 0.05. Note: Due to odds ratios having a minimum value of 0 (i.e., cannot be less than 0) and an infinite maximum value, the 95% confidence intervals were asymmetrical in magnitude.

**Table 1 ijerph-19-12225-t001:** Self-reported sample characteristics for 341 undergraduate students from Canada.

Participants (*n* = 341)	Mean ± SD [Range] or *n* (%)
Sex (Male, Female)	62 (19%), 279 (81%)
Gender (Men, Women, Non-binary)	64 (18%), 271 (79%), 6 (3%)
Age (years)	23 ± 4 [18, 40]
Body Mass Index (kg·m^−2^)	24.5 ± 5.0 [12.3, 42.2]
Racialized (Yes, No)	117 (33%), 224 (66%)
Varsity Athlete (Yes, No)	63 (18%), 278 (82%)
Year of Study	3 ± 1 [1, 7]
Academic Discipline (Health, Natural Sciences, Humanities)	233 (68%), 45 (13%), 56 (16%)
Location of School (West, Central, Atlantic)	103 (30%), 141 (41%), 97 (29%)
MVPA Levels (mins·week^−1^)	271 ± 297 [0, 1092]
Met Physical Activity Guidelines (Yes, No)	183 (53%), 158 (47%)
Sedentary Levels (h·day^−1^)	7.8 ± 2.7 [1.0, 16.6]
Met Sedentary Guidelines (Yes, No)	168 (49%), 173 (51%)

MVPA, moderate-to-vigorous-intensity physical activity. Note: Data are reported as Mean ± SD [Minimum, Maximum].

**Table 2 ijerph-19-12225-t002:** The frequency of each motivator and barrier to exercise among Canadian undergraduate students.

Motivator	*n* (%)	Barrier	*n* (%)
Improved Physical Health	254 (74%)	School Obligations	232 (68%)
Improved Mental Health	228 (67%)	Time Commitment	197 (58%)
Improved Physical Appearance	203 (60%)	Job Obligations	109 (32%)
Improved Athletic Performance	94 (28%)	Lack of Fitness Classes	88 (26%)
Reduced Risk of Disease	83 (24%)	Lack of Interest	60 (18%)
Personal Challenge	42 (12%)	Cost	58 (17%)
Self-Identity	42 (12%)	Self-Confidence Issues	55 (15%)
Academic Performance	38 (11%)	Lack of Facilities	49 (14%)
Socializing with Others	38 (11%)	Cannot Exercise Alone	48 (14%)
		Body Image Concerns	34 (10%)
		Family Obligations	34 (10%)
		Medical Conditions	25 (7%)
		Volunteerism	21 (6%)
		Research Obligations	13 (4%)

Note: *n* (%) represents the number of times each motivator/barrier was selected in the top 3 with % of the total sample in brackets. The top 4 selections were used in logistic regression analyses.

## Data Availability

All data files can be provided by the corresponding author, M.W.O., upon reasonable request.

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
