# Peer review of "The Influence of Motivators and Barriers to Exercise on Attaining Physical Activity and Sedentary Time Guidelines among Canadian Undergraduate Students"

_ijerph, 2022, doi:10.3390/ijerph191912225_

Round 1
Reviewer 1 Report
Authors tackle an interesting topic which is the effect of motivation on the time devoted to physical activity and sedentary behaviour. Moreover, they select a population (post-secondary / undergraduates) that is less usual than children or adolescents and it is convenient, considering university time as a sensible period for the physical activity drop-out. Authors stand the line on the time guidelines recommendations as is common on literature. Unfortunately, the manuscript has some important weaknesses that do not allow me to recommend it for publication.
First of all, there is a lack of profundity of the theoretical framework of motivation, so the introduction is mainly focused on physical activity and sedentary time recommendations and the risk of not attending them, while the complex framework of motivation is reduced to introduce potential facilitators and barriers according a brief volume of studies. It is convenient to incorporate some systematic reviews and metanalysis about motivation and introduce some of the main theories as self-determination theory, the planned behaviour or the transcontextual model.
With regards of the self-reported method to collect information, although it is a fair limitation well accepted in terms of research methods, the questions are not precise enough to gather strong information for the analysis.
Apart from this, it is not clear to this reviewer why participants only choose among nine motivators and fourteen barriers. Why these and no others? Why only the top three?
In terms of explaining the complexity of the motivational framework and its impact on physical activity and sedentary behaviours, why is more relevant that 28% of sample indicate as motivator the athletic performance than the 24% that state the risk of disease? What about the other facilitators and barriers? Is there nothing to say about social environment or body image concerns?
From my experience, a top 3 frequency and some logistic regressions do not allow to go in depth on the relevance of the topic. That is probably why the manuscript one has 10 pages and 19 references.
Finally, there are some other questions important to rethink for future submissions such as the incoherence of asking for sex (binary) and gender (not binary) or declaring a sample around four hundred participants as a large sample.
In sum, this version of the manuscript does not reach the level for considering the publication in IJERPH.
Reviewer 2 Report
Dear Authors,
The article requires a very clear description of its novelty and the need to research this particular population group. It is crucial to review some issues
Сan this sample be considered representative? What makes it possible to claim that the obtained results can be extrapolated to all Canadian students?
What was the number of students who withdrew from the study?
Kind regards,
Reviewer 3 Report
Dear Authors,
I revised the manuscript intitle: The Influence of Motivators and Barriers to Exercise on Attaining Physical Activity and Sedentary Time Guidelines Among Canadian Undergraduate Students.
This work presents the results of a national survey sent to Canadian undergraduate students aiming to determine motivators and barriers to exercise; evaluate the amount of students meeting physical activity and sedentary guidelines and identity which motivators or barriers influence most the behaviours to reach the guidelines.
The question asked is interesting and brings new evidence from the previous surveys cited in the manuscript.
The manuscript is well written and clearly understandable.
Major comments:
Introduction
I have a comment regarding the objective stated in the introduction. The authors wrote: "The purpose of this study was to determine the most common motivators and barriers to exercise among a diverse sample of undergraduate students. ". However, they clearly describe before that those motivators and barriers are already known. "While motivators and barriers to exercising in 55 post-secondary students have been investigated in individual Canadian colleges/universities [5,9] and the following paragraph.
What is new here is the analysis of their influence on guideline adherence.
I think the objective of the manuscript should focus more on this topic which is new rather than finding the motivators/barriers that are already known in the literature.
Methods
The first sentence of the method gives a result. This sentence should appear in the results section. The methods section should describe how the online survey was built and distributed and how the participants were selected, describing the inclusion and non-inclusion criteria. (Most of this is clearly described in the method section).
I have no further comment.
Minor comments:
please be careful with the typo, in a few places a double space is present. (i.e. p.1 L.40; p.2 l.49; )
Round 2
Reviewer 1 Report
Authors have incorporated some changes in the manuscript and argue the ones they respectfully disagree. However, for this reviewer, the approach, the research methods selected and the results of the paper do not allow to explain the complexity of the motivation.
Reviewer 2 Report
Dear Authors, thank you for the answer and all clarifications.
Best regards,
